# Adiponectin Enhances Fatty Acid Signaling in Human Taste Cells by Increasing Surface Expression of CD36

**DOI:** 10.3390/ijms24065801

**Published:** 2023-03-18

**Authors:** Fangjun Lin, Yan Liu, Trina Rudeski-Rohr, Naima Dahir, Ashley Calder, Timothy A. Gilbertson

**Affiliations:** 1Burnett School of Biomedical Sciences, College of Medicine, University of Central Florida, Orlando, FL 32827, USA; fangjun.lin@ucf.edu (F.L.); trudeski@knights.ucf.edu (T.R.-R.); ndahir@umich.edu (N.D.); ascalder@med.umich.edu (A.C.); 2Department of Internal Medicine, College of Medicine, University of Central Florida, Orlando, FL 32827, USA; liu.yan.7605@gmail.com

**Keywords:** adiponectin, AdipoRon, taste cells, fat taste, CD36

## Abstract

Adiponectin, a key metabolic hormone, is secreted into the circulation by fat cells where it enhances insulin sensitivity and stimulates glucose and fatty acid metabolism. Adiponectin receptors are highly expressed in the taste system; however, their effects and mechanisms of action in the modulation of gustatory function remain unclear. We utilized an immortalized human fungiform taste cell line (HuFF) to investigate the effect of AdipoRon, an adiponectin receptor agonist, on fatty acid-induced calcium responses. We showed that the fat taste receptors (CD36 and GPR120) and taste signaling molecules (Gα-gust, PLCβ2, and TRPM5) were expressed in HuFF cells. Calcium imaging studies showed that linoleic acid induced a dose-dependent calcium response in HuFF cells, and it was significantly reduced by the antagonists of CD36, GPR120, PLCβ2, and TRPM5. AdipoRon administration enhanced HuFF cell responses to fatty acids but not to a mixture of sweet, bitter, and umami tastants. This enhancement was inhibited by an irreversible CD36 antagonist and by an AMPK inhibitor but was not affected by a GPR120 antagonist. AdipoRon increased the phosphorylation of AMPK and the translocation of CD36 to the cell surface, which was eliminated by blocking AMPK. These results indicate that AdipoRon acts to increase cell surface CD36 in HuFF cells to selectively enhance their responses to fatty acids. This, in turn, is consistent with the ability of adiponectin receptor activity to alter taste cues associated with dietary fat intake.

## 1. Introduction

Obesity remains a major public health challenge worldwide and is labeled as a national epidemic in the US with up to 40% of adults having excessive body fat accumulation and body mass index ≥30 kg/m^2^ in 2020 [1]. Obesity decreases life expectancy and quality of life; it also increases a person’s risk to develop a number of preventable health conditions that contribute to chronic illness and death, including heart disease, stroke, hypertension, and type 2 diabetes mellitus. Lifestyle, particularly eating behavior, plays an important role in the development of obesity, and excess calorie consumption can result in disproportional energy intake and expenditure. Indeed, dietary fat intake is implicated in the development of obesity specifically by a positive correlation between fat intake and weight [2]. Foods with high-fat content are both calorically dense and palatable [3]. As the gustatory system plays a crucial role in promoting nutrient intake, a preference for the taste of fats is due, in part, to dietary fatty acids activating unique lipid-sensing chemoreceptors found on taste bud cells that are embedded in the tongue. However, increased consumption of dietary fat reduces our ability to detect fatty acids, therefore requiring larger amounts of fat in our diet to achieve pleasurable taste. Moreover, taste is weakened in obese, especially the taste for fat [4,5]. 

Adiponectin is secreted into the circulation by adipocytes in response to calorie restriction. A decline in circulating adiponectin levels is observed in obesity and has been suggested to play an important role in the pathogenesis of several obesity-related conditions including heart disease and type 2 diabetes mellitus [6,7]. Adiponectin exerts its pleiotropic functions by interaction with three key receptors including AdipoR1, AdipoR2, and T-cadherin [8,9]. These receptors have attracted great interest as potential therapeutic targets for multiple conditions including obesity, cardiovascular disease, and diabetes. AdipoRon, an orally active adiponectin receptor agonist, binds with high affinity to the adiponectin receptors AdipoR1 and AdipoR2, which acts via 5’ adenosine monophosphate-activated protein kinase (AMPK) and peroxisome proliferator-activated receptor alpha (PPARα) pathways, respectively [10]. Importantly, AdipoRon acts in a manner consistent with the activity of adiponectin in a number of physiological systems [11,12,13,14,15,16]. Recently, Crosson and colleagues found that adiponectin receptors are highly expressed in mice taste buds [17]. Therefore, it has been hypothesized that through its receptors, adiponectin has the potential to markedly affect peripheral taste signaling. Indeed, there are accumulating data demonstrating that circulating peptides/hormones act on their receptors, which are present in the peripheral gustatory system, to regulate taste function and preference. Additionally, behavioral studies showed that salivary gland-specific adiponectin rescue in adiponectin knockout mice significantly increases behavioral taste responses to intralipid (fat stimulus) [17]. However, the molecular and cellular mechanisms that underlie adiponectin’s role in fatty acid detection in the gustatory system remain unclear.

The transduction of polyunsaturated fatty acids (PUFAs) in the mammalian taste system has been extensively explored. Briefly, PUFAs such as linoleic acid, the most extensively studied fatty acid, activate GPR120 and CD36 to initiate a signaling pathway that involves, at least in part, the activation of G proteins and the production of the ß2 isoform of phospholipase C (PLC), the liberation of IP_3_ and diacylglycerol (DAG), the rise in intracellular free Ca^2+^ and activation of transient receptor potential channel subtypes M5 and M4 (TRPM5, TRPM4), culminating in eventual neurotransmitter release (for review, see [18,19,20]). Given the overlap between the transduction elements of fatty acid signaling and the expression of adiponectin receptors in the taste system, it is plausible that adiponectin might regulate the expression or activity of the molecular components of the fatty acid transduction pathway in the gustatory system. One potential target of adiponectin signaling is the fatty acid translocase cluster of differentiation 36 (CD36) which is known to facilitate fatty acid uptake in multiple tissue types and also serves as a detector for dietary fatty acids in the taste system. Interestingly, in rodents adiponectin upregulates CD36 expression [21] and increases the translocation of CD36 to the plasma membrane via the activation of AMPK [22]. Following adiponectin’s demonstrated regulatory effects on the uptake of long-chain fatty acids [23,24,25], we hypothesized that in the taste receptor cells of the peripheral gustatory system, adiponectin may similarly modulate fatty acid responsiveness.

Taste transduction pathways are well characterized in rodent models, but studies in humans have been comparatively rare. Recent attempts to develop human taste cell lines hold promise for expanding these mechanistic studies to further understand taste transduction pathways in human cells. In the present study, we utilized a recently developed immortalized human fungiform taste cell line (HuFF) to identify adiponectin’s role, if any, in fatty acid-induced cellular responses. Specifically, we investigated the potential effect of the adiponectin receptor agonist, AdipoRon, on fatty acid-induced calcium responses in HuFF cells. The aims of this study were (1) to validate if the HuFF cells are functionally comparable to primary rodent taste cells and can serve as a model for studying fatty acid taste signaling, (2) to determine whether AdipoRon selectively enhances the cellular responses to fatty acids in HuFF cells, and (3) if so, to examine whether AdipoRon enhances fatty acids responses via AMPK activation and subsequent CD36 translocation to the plasma membrane.

## 2. Results

### 2.1. HuFF Cells Express Adiponectin Receptors and Fat Taste Receptors

Previous studies have shown that fat taste receptors (CD36 and GPR120) are expressed in human fungiform taste bud cells [26] and that CD36 is co-expressed with GPR120, PLCβ2, and Gα-gust [27]. Here, we used quantitative RT-PCR and immunofluorescence assay to determine the expression of adiponectin receptors (AdipoR1, AdipoR2, and T-cadherin), fat taste receptors (CD36 and GPR120), and downstream taste signaling molecules (PLCβ, Gα-gust, and TRPM5). As a result, mRNAs for *AdipoR1*, *AdipoR2*, *CDH13*, *CD36*, *GPR120*, *PLCβ2*, and *GNAT3* were expressed in HuFF cells (Figure 1). Immunofluorescence microscopy also confirms that AdipoR1, CD36, GPR120, PLCβ2, Gα-gust, and TRPM5 expression was visible in HuFF cells (Figure 2) and that AdipoR1 is co-expressed with CD36 (Figure 2A) and GPR120 (Figure 2B).

### 2.2. HuFF Cells Differentially Respond to Bitter, Sweet, Umami, and Fatty Acids

To determine if the HuFF cells are functionally comparable to primary rodent taste cells, calcium imaging studies were performed to explore the cellular responsiveness to different taste stimuli. As shown in Figure 3A, saccharin (20 mM; “sweet”), denatonium benzoate (DB, 5 mM; “bitter”), linoleic acid (LA, C18:2, 30 µM; “fatty acid”), and capric acid (C10:0, 100 µM; “saturated fatty acid”) elicited reversible calcium responses in HuFF cells, but monosodium glutamate (MSG, 20 mM) did not reliably generate any changes in intracellular calcium in a significant proportion of HuFF cells (only 4 responding cells in a total of 280 cells; 1.4%). The overlap in the HuFF cells’ responses to saccharin (27.8% of cells responding), DB (18.9%), LA (58.6%), and capric acid (7.5%) are presented in Appendix A (*n* = 280 total cells). 

In rodent taste cells, long-chain fatty acid responses are limited to *cis*-unsaturated fatty acids [18,28]. To determine the type of fatty acid that could elicit cellular responses in HuFF cells, various cis-PUFAs (docosahexaenoic acid, DHA, C22:6; eicosapentaenoic acid, EPA, C20:5; arachidonic acid, AA, C20:4), monounsaturated fatty acids (MUFAs; nervonic acid, NA, C24:1; erucic acid, EA, C22:1), and one trans-PUFA (linolelaidic acid, LEA, *trans-*C18:2) were applied during calcium imaging. As shown in Figure 3B, HuFF cells only responded to *cis*-PUFAs (DHA, 22 in 46 cells; AA, 29 in 46 cells; EPA, 35 in 46 cells), and did not reliably respond to trans-PUFAs (LEA, 1 in 46 cells) or MUFAs (NA, 2 in 46 cells; EA 3 in 46 cells).

### 2.3. HuFF Cells Act as a Model System for Exploring Fatty Acid Signaling

To determine if HuFF cells were able to recapitulate the transduction pathway for PUFAs that has been well established in mouse primary taste cells, we performed a series of experiments using ratiometric calcium imaging focusing on response to linoleic acid, the prototypical PUFA stimulus. Similar to responses shown in rodents [29,30], HuFF cells showed a robust rise in intracellular calcium in a dose-dependent manner in response to a series of LA concentrations ranging from 10 to 200 µM (Figure 4A). To compare across cells from different preparations, the area under the curve for each response was determined and normalized relative to the response to 30 µM LA in the same cell. Data from 27–50 cells per point were averaged and fit with a logistic relation to determine a relative EC_50_ for LA of 64.03 µM (Figure 4A).

Generally, there are two receptors, GPR120 and CD36, that have been most commonly implicated in the initial transduction of PUFAs in the taste system as mentioned above. To confirm the involvement of CD36 and GPR120 in the LA-induced signaling pathway in HuFF cells (Figure 4B–D), we conducted calcium imaging using a series of pharmacological agents targeting specific elements in the transduction pathway. As expected, the intracellular calcium responses induced by 30 µM LA were significantly reduced by the irreversible CD36 inhibitor, sulfosuccinimidyl oleate (SSO, 400 µM, 20 min pre-treatment, 41.3 ± 2.8% [SEM] reduction; Figure 4B), and the GPR120 antagonist AH-7614 (10 µM, 20 min pre-treatment, 56.4 ± 3.7% reduction; Figure 4B), consistent with the contribution from these two receptors. The contribution of other pathway elements (cf. [20]) was also verified using this same approach. Treatment with triphenylphosphine oxide (TPPO, a reversible and selective blocker for the TRPM5 channel, 200 µM, 78.4 ± 2.7% reduction; Figure 4B,C), and U73122 (PLC inhibitor, 6 µM, 66.7 ± 5.4% reduction; Figure 4B,D) inhibited LA-induced intracellular calcium responses in HuFF cells. In addition, LA-induced calcium responses were inhibited by a store-operated calcium entry (SOCE) blocker BTP2 (10 µM, 2 h pre-treatment, 88.4 ± 2.5% reduction; Figure 4B). All inhibitions by pharmacological agents were significant (*p* < 0.001) as assessed by independent one-sample *t*-tests (Figure 4B).

### 2.4. AdipoRon Selectively Enhances Calcium Responses to Fatty Acids in HuFF Cells

AdipoRon is a selective agonist for both AdipoR1 and AdipoR2 with dissociation constants (Kd) of 1.8 μM and 3.1 μM, respectively [10]. AdipoRon was demonstrated to increase intracellular calcium levels in high-glucose-treated human glomerular endothelial cells and murine podocytes [31]. In contrast to those cells, however, AdipoRon (5 µM) alone did not reliably generate any measurable changes in intracellular calcium levels in HuFF cells (Figure 5A). While LA (30 µM) alone reliably induced calcium responses of 104.39 ± 8.54 nM (*n* = 436), the addition of AdipoRon increased the magnitude of the LA-induced calcium responses in a concentration-dependent manner (0.1 µM: 89.81 ± 7.95, *n* = 109; 0.5 µM: 123.11 ± 23.27, *n* = 110; 1 µM: 135.47 ± 17.63, *n* = 97; 5 µM: 225.30 ± 29.00, *n* = 159; and 10 µM: 250.45 ± 29.47, *n* = 121, Figure 5B,C). The EC_50_ for the enhancement effect of AdipoRon on calcium responses to LA was 1.67 µM (Figure 5D), which was close to the previously reported Kd (1.8 μM) of AdipoRon bound to AdipoR1. In marked contrast, calcium responses to a mixture of sweet (20 mM saccharin), bitter (3 mM denatonium benzoate, DB, and 0.1 mM cycloheximide), and umami (5 mM monosodium glutamate, MSG) stimuli were not affected by AdipoRon treatment (t_(100)_ = 0.011, *p* = 0.9912, unpaired *t*-test, Figure 5E,F), indicating that AdipoRon selectively enhances calcium responses to fatty acids in HuFF cells. 

### 2.5. AdipoRon Acts on CD36 to Increase Fatty Acids-Induced Responses in HuFF Cells

Since AdipoRon selectively increased the LA-induced calcium responses in HuFF cells, we next sought to determine which of the fatty acid signaling pathway is involved in AdipoRon-modulated fatty acids responses in HuFF cells. A series of calcium imaging experiments were performed to test whether there are any changes in the enhancement effect of AdipoRon on fatty acid responses of HuFF cells by pharmacologically blocking the activation of fatty acid receptors. The results showed that the CD36 antagonist SSO inhibited (t_(169)_ = 2.149, *p* = 0.0331, unpaired *t*-test, Figure 6A,C) and AdipoRon enhanced (t_(136)_ = 2.879, *p* = 0.0046, unpaired *t*-test, Figure 6B,C) the LA-induced calcium responses. Blocking CD36 eliminated the ability of AdipoRon to enhance LA-induced calcium responses (t_(203)_ = 1.654, *p* = 0.0996, unpaired *t*-test, Figure 6A,C). In contrast, AH-7614 (GPR120 antagonist) inhibited LA-induced calcium responses (t_(247)_ = 2.366, *p* = 0.0187, unpaired *t*-test, Figure 6D–F), but did not affect the ability of AdipoRon to increase LA-induced responses (t_(238)_ = 4.487, *p* < 0.0001, unpaired *t*-test, Figure 6D,F). In addition, AdipoRon had no effect on GW9508 (an agonist for GPR120 [and GPR40])-induced calcium responses in HuFF cells (t_(90)_ = 0.2709, *p* = 0.7871, unpaired *t*-test, Figure 6F). 

Similar to its effects on LA responses, inhibition of CD36 inhibited cellular responses of HuFF cells to EPA (t_(125)_ = 3.388, *p* = 0.0009, unpaired *t*-test, Appendix A) and DHA (t_(144)_ = 2.017, *p* = 0.0456, unpaired *t*-test, Appendix A). Further, AdipoRon increased the calcium responses of HuFF cells to EPA (t_(146)_ = 2.305, *p* = 0.0226, unpaired *t*-test, Appendix A) and DHA (t_(168)_ = 2.296, *p* = 0.0229, unpaired *t*-test, Appendix A), and the effect of AdipoRon on these fatty acid-induced calcium responses was dependent upon the activity of CD36 since inhibition of CD36 with SSO blocked any enhancement (EPA, t_(104)_ = 0.0195, *p* = 0.9845; DHA, t_(120)_ = 1.760, *p* = 0.0810, unpaired *t*-test, Appendix A). Interestingly, however, another PUFA, arachidonic acid (AA) did not show a similar result. AA induced a rise in intracellular calcium that was not affected by the application of SSO (t_(96)_ = 0.3238, *p* = 0.7468, unpaired *t*-test, Appendix A) and AH-7614 (t_(110)_ = 0.08296, *p* = 0.9340, unpaired *t*-test, Appendix A). In addition, AdipoRon did not enhance the calcium responses of HuFF cells to AA (t_(100)_ = 0.2175, *p* = 0.8282, unpaired *t*-test, Appendix A). LA has been reported to induce the production of AA that will open the Orai1/3 channels, which are responsible for the SOCE in mice taste bud cells [32]. Consistent with this finding, the SOCE blocker BTP2 inhibited AA-induced calcium responses in HuFF cells (t_(169)_ = 7.786, *p* < 0. 0001, unpaired *t*-test, Appendix A). Taken together, these results indicate that CD36, but not the GPR120, may be responsible for the ability of AdipoRon to enhance fatty acid-induced responses in HuFF cells. 

### 2.6. AdipoRon’s Effect on LA-Induced Responses Is Mediated by Activation of AMPK

It has been shown in numerous studies that phosphorylation of AMP-activated protein kinase (AMPK) mediates many of the effects of AdipoRon (adiponectin) at the cellular level in a variety of cell types [33,34,35]. To test for the contribution of AMPK in the HuFF cell PUFA responses, we performed an ELISA assay for phosphorylated AMPKα using a commercially available kit. Our results show a significant (1.3-fold) increase in the phosphorylation state of AMPKα (Thr^172^) under the stimulation of AdipoRon for 30 min, which was inhibited by an AMPK inhibitor, dorsomorphin (compound C; F_(2,6)_ = 24.38, *p* = 0.0013, Figure 7A). To link the activation of AMPK to the enhancement role of AdipoRon on fatty acid-induced calcium responses, we examined the effect of compound C on AdipoRon’s effect on LA-induced calcium responses in HuFF cells. A one-way ANOVA showed that the ability of AdipoRon to enhance LA-induced calcium responses in HuFF cells was significantly inhibited by the treatment of compound C (F_(2,189)_ = 14.28, *p* < 0.0001, Figure 7B,C).

### 2.7. AdipoRon-Stimulated CD36 Translocation Is Dependent on AMPK Activation 

Previous studies demonstrated that adiponectin increases CD36 translocation to the plasma membrane in cardiomyocytes and upregulates CD36 expression in L6 myotubes via activation of AMPK [21,22]. Therefore, to test if AdipoRon similarly affects CD36 translocation in HuFF cells via AMPK, we monitored the subcellular distribution of CD36 in AdipoRon-treated HuFF cells using an immunofluorescence assay (Figure 8A) and a CD36 translocation assay (Figure 8B). After a 1 h serum starvation, little CD36 could be detected on the cell surface (and was likely localized to the perinuclear region) in the control group (Figure 8A). By contrast, with stimulation by AdipoRon (5 μM for 5 or 30 min), CD36 was recruited to the cell surface (Figure 8A), which resulted in an approximately 1.4-fold increase in cell surface CD36 (5 min: F_(3,16)_ = 40.14, *p* < 0.0001; 30 min: F_(3,16)_ = 8.411, *p* = 0.0014, Figure 8B). The change in CD36 expression caused by AdipoRon was not due significantly to an increase in gene expression. Using real-time PCR assays on mRNA isolated from HuFF cells treated with AdipoRon in the presence and absence of compound C, we measured the relative expression of *CD36*, *AdipoR1*, *AdipoR2*, and *CDH13* (T-cadherin). AdipoRon application did not change the relative expression of *AdipoR1*, *AdipoR2*, *CDH13*, and *CD36* in HuFF cells (Figure 8C). Therefore, AdipoRon apparently increases the surface expression of CD36 via AMPK activation but does not alter CD36 transcription in HuFF cells.

## 3. Discussion

The continuing high incidence of obesity is a major public health challenge worldwide. Obesity increases the health risks associated with many chronic morbidities, such as diabetes, cardiovascular disease, metabolic disease, and cancer [36,37]. Lifestyle choices, particularly eating behavior coupled with a sedentary lifestyle, are considered major factors in weight gain [38,39]. Taste is a key driver in food selection and may enhance preference for foods high in fat and sugar [40], which are both palatable and energy dense. However, high dietary fat reduces hedonic responses and sensitivity to fat taste [41], which may increase the drive for fat consumption, leading to chronic positive energy balance, and resulting in obesity. Moreover, obese subjects present a diminished ability to detect fatty acids with a much stronger preference for fat-rich foods [42,43] indicative of an inverse relationship between peripheral fat sensitivity and overall intake, which further increases the risk of adiposity. By contrast, fat restriction in obese people increases fat taste sensitivity [5], and individuals with high-fat sensitivity present lower consumption and less preference towards foods that are high in fat [4,44]. Thus, targeting the taste of fat might be a potential therapeutic approach for the management of fat intake and body weight.

Dietary fats are critical for our life and health. For example, essential fatty acids (which cannot be produced by metabolic processes in humans) must be obtained from food. Therefore, the ability to detect these essential fatty acids in food sources is necessary for survival. However, the oral detection of dietary fat was initially thought to be dependent on its texture rather than its taste. Our group provided some of the first evidence for the ability of fatty acids to activate our taste cells and has worked on exploring underlying mechanisms our body uses to recognize and respond to dietary fat [28]. Since then, accumulating evidence from humans and other animals provided support that fat may be classified as the sixth basic taste [45,46,47]. Indeed, long-chain fatty acids have been reported to elicit a unique taste sensation in humans [45]. Both CD36 and GPR120 are thought to be the functional fat taste receptors in taste buds cells [48,49]. The downstream transduction signaling pathways are well characterized in rodent models. In brief, long-chain fatty acids bind to CD36 to induce the activation of Src-PTKs [50] or bind to GPR120 causing the release of G proteins which, in turn, stimulate PLC and generate inositol 1,4,5-phosphate (IP_3_), resulting in an elevation of intracellular calcium and the opening of TRPM5 channels that are responsible for taste cell depolarization [20]. Human taste bud cells express fat taste receptors (CD36 and GPR120) and downstream signaling elements (Gα-gust and PLCβ), and in response to fatty acids show an increase in the intracellular calcium level [27]. However, in contrast to rodents, much less is known about fat taste transduction pathways in human taste cells. In this study, we show the expression of fat taste signaling elements and calcium responses in an immortalized human fungiform taste cell line (HuFF). Dose-dependent calcium responses to LA in HuFF cells were similar to the range of effective concentrations seen in rodent taste cells, and LA-induced calcium responses were significantly reduced by the administration of CD36, GPR120, PLCβ, and TRPM5 inhibitors. These data suggest that HuFF cells are functionally comparable to primary rodent taste cells and may serve as an appropriate model for studying fatty acid taste signaling in humans. While we have focused on their role in fatty acid signaling HuFF cells could serve as a model to study sweet and bitter signal transduction as well. Approximately 30% and 20% of HuFF cells responded to saccharin and DB, respectively. Although umami taste shares a similar downstream signal transduction pathway with the fat, sweet, and bitter taste, HuFF cells do not appear to be a good model for investigating umami taste signaling. It was beyond the scope of this research to fully characterize HuFF cells and a caveat is that we did not investigate a range of concentrations for other tastants in the present study.

Recently, a number of studies have shown that many peptide hormones and their receptors, such as Peptide YY, glucagon-like peptide-1 (GLP-1), leptin, cannabinoid, ghrelin, estrogen, and adiponectin that are classically considered to regulate food intake, also are present in taste bud cells and play a direct role in the modulation of fat taste responsiveness. Peptide YY gene knockout (PYY^−/−^) mice displayed a reduction in behavioral responsiveness to fat emulsions and it was effectively rescued by the reconstitution of salivary PYY [51]. Intraperitoneal injection of Exendin-4 (GLP-1 receptor agonist) reduced the lick responses and trial initiation of rats to both intralipid and sucrose during brief-access tests [52]. Leptin inhibited LA-induced intracellular calcium responses in taste bud cells and decreased the gustatory preference for LA in mice, whereas gene silencing of leptin or its receptor via the application of siRNAs onto the mice’s tongues upregulated the LA preference [53]. Cannabinoid 1 receptor gene knockout (CB_1_R^−/−^) mice showed a lower preference for fatty solutions compared to the WT controls, while LA-induced calcium responses were decreased in taste cells by pharmacologically (rimonabant, a specific CB_1_R inverse agonist) or genetically (CB_1_R^−/−^ mice) blocking the function of CB_1_R [54]. Ghrelin knockout (ghrelin^−/−^) and ghrelin O-acyltransferase knockout (GOAT^−/−^) mice demonstrated reduced expression levels of fat taste receptors (CD36 and GPR120) in taste bud cells and exhibited decreased fat taste sensitivity, compared to WT mice [55]. We also found a reduction of fat responsiveness in female ghrelin receptor knockout mice following 6 weeks of a 60% high-fat diet, but not in males, compared to WT controls [29]. Fatty acid-induced taste bud cell activation and fat taste sensitivity in mice were reported to be increased by estrogen, which was considered a major contributor for the sex differences in fat taste [30]. It has been found that adiponectin receptors are highly expressed in taste buds and salivary gland-specific adiponectin rescue in adiponectin knock out mice significantly increases behavioral taste responses to intralipid [17]. Interestingly, a similar positively correlated link between adiponectin level and fat taste sensitivity is also found in different genders. The adiponectin levels in both saliva and plasma are higher in females than in males coupled with a greater taste sensitivity to fatty acids [30,56,57]. Taken together, these findings provided evidence for the hormonal involvement in fat taste perception; however, there is still little understanding of the molecular and cellular mechanisms underlying hormonal peptide regulation in the taste system. In this study, we show the target of fat taste modulation by the adiponectin receptor agonist, AdipoRon, in HuFF cells. Our results showed that AdipoRon selectively enhances fatty acids-induced calcium responses via modulation the cell translocation of CD36, and this enhancement role of AdipoRon on fat taste is dependent upon the activation of 5’ adenosine monophosphate-activated protein kinase (AMPK).

Adiponectin is a hormone primarily secreted from adipose tissue, with the monomeric protein post-translationally modified into different molecular weight multimers: trimer (low), hexamer (middle), and 12–18 monomers (high) [58]. There are three proteins (AdipoR1, AdipoR2, and T-cadherin) that have been identified as adiponectin receptors that mediated the pleiotropic actions of adiponectin [8,9]. AdipoR1 has a high affinity for globular adiponectin, while AdipoR2 displays an intermediate affinity for both globular and full-length adiponectin, whereas T-cadherin shows a high affinity for the middle and high molecular weight multimers [8,9]. AdipoR1 and AdipoR2 are structurally related to each other and ubiquitously expressed in many tissues, while T-cadherin is structurally different from AdipoRs and there is comparatively little known about its signaling pathway. Activation of AdipoR1/APPL1 by adiponectin increases CD36 translocation and fatty acid uptake via the phosphorylation of AMPK in rat cardiomyocytes [22]. Our qRT-PCR results were consistent with the high expression of adiponectin receptors in taste cells [59], and the double-labeled immunostaining showed that AdipoR1 was co-expressed with fat taste receptors (CD36 and GPR120). These results provide insights into the role of adiponectin signaling in modulating fat taste. To test this possibility, we examined the potential effects of the adiponectin receptor agonist AdipoRon on intracellular calcium responses to LA as well as a taste mixture of sweet, bitter, and umami in HuFF cells. Unlike the high-glucose-treated human glomerular endothelial cells and murine podocytes [31], AdipoRon (5 µM) alone did not alter the intracellular calcium levels in HuFF cells. We found that AdipoRon enhances the LA-induced calcium responses in a dose-dependent manner with an EC_50_ value of 1.67 µM, which was close to that previously reported Kds of AdipoRon bound to AdipoR1 (1.8 μM) and AdipoR2 (3.1 μM) [10]. RNA interference or CRISPR gene knockout techniques are needed in future studies to address which of the adiponectin receptors mediates the role in fat taste. A previous study showed that no significant difference in behavioral taste responses has been found between adiponectin KO and WT mice, but salivary gland-specific adiponectin rescue in adiponectin KO mice significantly increased the brief-access taste responses to intralipid stimulus, but not for sucrose and QHCl [17]. Consistent with this behavioral study, our calcium imaging data showed that AdipoRon enhances the cellular responses of HuFF cells to LA, but not for the sweet, bitter, and umami mixture. 

Next, we provided evidence that the CD36 pathway, independent of GPR120, is functionally responsible for the enhancement role of AdipoRon on fatty acids responses. Pharmacologically blocking the function of CD36 by SSO eliminated the enhancement effect of AdipoRon on fatty acid-induced responses (LA, EPA, and DHA). Similar results in HL-1 cell studies have also shown that SSO completely prevented insulin-stimulated fatty acid uptake [60]. In contrast, blocking the function of GPR120 by AH-7614, AdipoRon was still able to enhance the LA-induced calcium responses. Moreover, GPR120 agonist GW9508-induced calcium response was not affected by the application of AdipoRon. AA acts on different types of ion channels, such as TRP channels, SOCE, and non-SOCE channels, and plays a variety of functions in living cells. In mice taste bud cells, LA has been shown to induce the production of AA that will open the Orai1/3 channels, which are responsible for the SOCE [32]. Interestingly, we also found that AdipoRon did not enhance the calcium responses induced by AA in HuFF cells. Further studies showed that BTP2, but not SSO and AH-7614, inhibited AA-induced calcium responses in HuFF cells, suggesting that AA may act directly on SOCE channels to induce calcium influx independent of either CD36 or GPR120. These data suggest that AdipoRon would modify the fat taste responses of HuFF cells via the mediation of the CD36 pathway in these cells.

In the present study, we revealed the importance of AMPK in AdipoRon’s effect on fat taste. AdipoRon increased the phosphorylation of AMPKα (Thr^172^) and it was inhibited by AMPK inhibitor compound C. Similarly, the enhancement role of AdipoRon on LA-induced calcium responses of HuFF cells was significantly inhibited by the treatment of compound C. Studies in intestinal epithelial cells have suggested that AMPK is critical for CD36 translocation and long-chain fatty acid uptake [61]. CD36 dynamically traffics between the plasma membrane and subcellular compartments. Several studies have revealed that muscle contractions [62,63] and hormones, such as adiponectin [22] and insulin [64], could be the important factors that initiate the translocation of CD36 from intracellular compartments to cell surface membranes. Therefore, we hypothesize that AdipoRon selectively enhances the cellular response to fatty acids through the AMPK pathway, which increases the translocation of CD36 to the plasma membrane of HuFF cells. However, little is known about the time course dynamics of the CD36 translocation under stimulation. The enhancement of the LA-induced calcium responses by AdipoRon was rapid and could be seen within a few minutes of administration. The results from our present study suggested that the membrane recruitment of CD36 induced by AdipoRon is rapid and may occur in a few minutes, which could possibly explain why it selectively enhances taste responses to fatty acids. In support of these findings, it has been reported that 15 and 30 min application of adiponectin increased CD36 translocation from intracellular to the cell surface [22], and the translocation of CD36 was observed even within 1 min of muscle contractions [63]. Although the mechanisms initiating the translocation of CD36 are unclear, previous studies have demonstrated that CD36 translocation is stimulated by adiponectin through the activation of AMPK [22], by insulin via the PI3K/AKT signaling [65], and by muscle contractions in both AMPK-dependent and AMPK-independent manners [62,63]. Our results in this study indicated that AMPK is essential in the regulation of CD36 translocation induced by AdipoRon. Together, our results suggest that AdipoRon via the activation of AMPK promotes the cell surface translocation of CD36 and therefore enhances cellular responses to fatty acids. Future studies are needed to understand in greater detail adiponectin’s effects on an animal’s fat taste behavior.

## 4. Materials and Methods

### 4.1. Cell Culture

The commercially available human fungiform taste cell line (HuFF; Applied Biological Materials, Richmond, BC, Canada; Catalog #: T0029) was immortalized via serial passaging and transformation with recombinant lentiviruses carrying simian virus 40 Large T antigen. HuFF cells were cultured in Prigrow V medium supplemented with 10% fetal bovine serum and 1% penicillin/streptomycin solution under a humidified atmosphere containing 5% carbon dioxide at 37 °C. The cells were seeded on glass coverslips 4–24 h before calcium imaging and 24–48 h before immunofluorescence assay. HuFF cells cultured in 96-well plates and 6-well plates were used for the CD36 translocation assay and ELISA experiments, respectively.

### 4.2. Solutions

Standard Tyrode’s solution contained 140 mM NaCl, 5 mM KCl, 1 mM CaCl_2_, 1 mM MgCl_2_, 10 mM HEPES, 10 mM glucose, and 10 mM Na pyruvate; adjusting the pH to 7.40 with NaOH; 300–320 mOsm. Stock solutions of polyunsaturated fatty acids (Sigma-Aldrich, St. Louis, MO, USA or Cayman Chemical, Ann Arbor, MI, USA) were made in 100% ethanol and stored under nitrogen at −20 °C. All working solutions of the polyunsaturated fatty acids were made from stock solutions immediately before use. Taste mixture solution contained 115 mM NaCl, 5 mM KCl, 1 mM CaCl_2_, 1 mM MgCl_2_, 10 mM HEPES, 10 mM glucose, 10 mM Na pyruvate, 20 mM Na saccharin, 5 mM monosodium glutamate (MSG), 3 mM denatonium benzoate (DB), and 0.1 mM cycloheximide; adjusting the pH to 7.40 with NaOH at an osmolarity of 300–320 mOsm. Stock solutions of AH-7614 (Sigma-Aldrich), sulfosuccinimidyl oleate (SSO; Cayman Chemical), N-[4-[3,5-Bis(trifluoromethyl)-1H-pyrazol-1-yl]phenyl]-4-methyl-1,2,3-thiadiazole-5-carboxamide (BTP2; EMD Millipore), triphenylphosphine oxide (TPPO; Sigma-Aldrich), U73122 (Cayman Chemical), AdipoRon (MedChem Express, Monmouth Junction, NJ, USA), GW9508 (MedChem Express), and dorsomorphin (compound C; ApexBio, Houston, TX, USA) were made in DMSO and diluted the day of the experiment to a designated concentration with Tyrode’s.

### 4.3. Quantitative RT-PCR Analysis 

HuFF cells cultured in surface-treated sterile tissue culture flasks (12.5 cm^2^, Fisher Scientific) were used for studies involving the measurement of gene expression of adiponectin receptors and fat taste signaling elements. To test the effect of AdipoRon on the expression of CD36 and adiponectin receptors (cf. Figure 8C), HuFF cells were incubated in 6-well plates with DMSO (control), 1 µM AdipoRon, 5 µM AdipoRon, and 5 µM AdipoRon plus 10 µM compound C for 20–24 h. Total RNA extracted from HuFF cells according to the RNAzol RT protocol (Molecular Research Center, Cincinnati, OH, USA), followed by the RNA clean and concentrator-25 kits (Zymo Research, Irvine, CA, USA) included an in-column DNase treatment. RNA integrity and purity were evaluated by using agarose gel electrophoresis and Nanodrop 8000 Spectrophotometer (Thermo Fisher Scientific, Waltham, MA, USA), respectively. Then, RNA samples were converted to cDNA by qScript cDNA Synthesis Kit (Quanta Biosciences, Beverly, MA, USA). Commercially available TaqMan assays (FAM-labeled), obtained from Fisher scientific: *AdipoR1* (Hs00360422_m1), *AdipoR2* (Hs00226105_m1), *CDH13* (Hs01004531_m1), *CD36* (Hs00354519_m1), *GPR120* (Hs00699184_m1), *GNAT3* (Hs01385398_m1), *PLCβ2* (Hs01080541_m1), *TRPM5* (Hs00175822_m1), were used to detect the gene expression of adiponectin receptors and fat taste signaling elements. The *GAPDH* qPCR probe assay (Hs.PT.39a.22214836, HEX-labeled, Integrated DNA Technologies, Coralville, IA, USA) was used as an internal control. Final reaction cocktail (20 µL) contained the following: 10 µL TaqMan master mix (2×), 1 µL Taqman assays (20×), 1 µL GAPDH probe (20×), 1 µL template, and 7 µL nuclease-free water. Quantitative real-time PCR analyses were carried out following TaqMan protocols according to the manufacturer’s instructions for the QuantStudio 3 Real-Time PCR System (Thermo Fisher Scientific). Four independent experiments, each consisting of three replicates, were conducted. Following the RT-PCR reaction, gene expression was quantified by measuring the cycle threshold (C_T_). Each target gene was compared with GAPDH and relative expression was normalized to the expression level for *CD36*, which served as the calibrator. The standard ΔΔC_T_ method was used to generate measures of relative gene expression for all targets of interest [30]. The mean relative expression ± SEM of each target gene was then calculated using the 2^−ΔΔCT^ analytical method [66]. 

### 4.4. Immunofluorescence Assay

HuFF cells, seeded on 15 mm glass coverslips in 12-well plates without any treatment for 1–2 days, were used for immunostaining experiments to determine the expression of AdipoR1 and fat taste signaling components. To monitor the subcellular distribution of CD36 under AdipoRon treatment, HuFF cells need to be washed and incubated in a serum-free medium for at least 1 h prior to the treatment. Then, serum-free medium with DMSO (control), 1 µM AdipoRon, 5 µM AdipoRon, and 5 µM AdipoRon plus 10 µM compound C were applied to the HuFF cells for 5 and 30 min.

The cells were fixed with cold 100% methanol for 10 min at 4 °C and washed three times (5 min for each) with cold PBS. Then, samples were placed in a blocking buffer (1% bovine serum albumin, 0.1% Tween-20 in PBS) for 1 h at room temperature and incubated with primary antibodies overnight at 4 °C: rabbit polyclonal anti-AdipoR1 (1:50; Invitrogen, Waltham, MA, USA), mouse monoclonal anti-CD36 (1:50; Abcam, Boston, MA, USA), rabbit polyclonal anti-CD36 (1:50; Santa Cruz Biotechnology, Dallas, TX, USA), mouse monoclonal anti-GPR120 (1:50; Santa Cruz Biotechnology), mouse monoclonal anti-PLCβ2 (1:50; Santa Cruz Biotechnology), rabbit polyclonal anti-Gα-gust (1:50; Santa Cruz Biotechnology), rabbit polyclonal anti-TRPM5 (1:100; Alomone Labs). After washing three times with PBS, the coverslips were incubated with goat anti-rabbit Alexa Fluor 594 (1:500, Invitrogen) and/or goat anti-mouse AlexaFluor488 (1:500, Invitrogen) for 1.5 h at room temperature. Subsequently, the cells were washed twice with PBS and counterstained with DAPI (1 µg/mL in PBS; Invitrogen) for 5 min at room temperature. Finally, coverslips were mounted to glass slides with Fluoromount G (Southern Biotech, Birmingham, AL, USA) and sealed using nail polish. The immunofluorescence images of the labeled HuFF cells were obtained using an all-in-one fluorescence microscope (BZX800, Keyence, Itasca, IL, USA).

### 4.5. Calcium Imaging 

HuFF cells were seeded on coverslips for at least 4 h, then loaded with 4 µM of Fura-2AM (Invitrogen) in Tyrode’s with 0.05% pluronic acid (Invitrogen) for 1 h in the dark. The coverslips were placed onto the perfusion chamber (RC-25F, Warner Instruments, Holliston, MA, USA). Tastant solutions were perfused extracellularly at a flow rate of 4 mL/min, followed by 1 min of 0.1% fatty acid-free BSA solution, and then regular Tyrode’s (about 2 min) until the calcium signal returned to near baseline level. Cells were illuminated with Lambda DG-5 (Sutter Instruments, Novato, CA, USA) or CoolLED pE-340fura (CoolLED, Andover, UK) illumination system, and imaging was performed using an acA720 camera (Basler, Ahrensburg, Germany) coupled to a microscope (Olympus CKX53). The cell fluorescence at excitation wavelengths of 340 and 380 nm was recorded at a rate of 20 pairs per minute and converted to calcium concentration according to a standard curve generated from the calcium calibration kit (Invitrogen) by InCyt Im2™ imaging software (Version 6.00, Cincinnati, OH, USA). 

### 4.6. AMPKα [pT172] ELISA

HuFF cells grown in 6-well plates were serum-starved for at least 1 h and then treated with DMSO (control), 1 µM AdipoRon, 5 µM AdipoRon, and 5 µM AdipoRon plus 10 µM compound C for 30 min. The cells were washed with cold PBS and collected by gentle scraping from the plate. Next, the cell pellet was lysed in RIPA lysis buffer (Thermo Fisher Scientific) with inhibitors (Thermo Fisher Scientific) for 30 min on ice and vortexed at 10 min intervals. The supernatant was collected after a 10 min centrifuge at 13,000 rpm. A bicinchoninic acid (BCA) assay was used for the total protein quantification following the manufacturer’s instructions of Pierce™ BCA protein assay kit (Thermo Fisher Scientific). A sample of 10 µg of total protein from each sample was used to determine the phosphorylation of AMPK, following the simple step-by-step protocols of the AMPKα [pT172] ELISA kit (Invitrogen).

### 4.7. CD36 Translocation Assay

HuFF cells were grown in 96-well plates until about 90% confluent and serum-starved for at least 1 h prior to treatment with DMSO (control), 1 µM AdipoRon, 5 µM AdipoRon, and 5 µM AdipoRon plus 10 µM compound C for 5 and 30 min. The cells were fixed with 3% paraformaldehyde for 10 min and blocked with blocking buffer (5% goat serum, 1% bovine serum albumin, 0.05% Tween-20 in PBS) for 1 h. Following incubation with rabbit polyclonal anti-CD36 antibody for 2 h, the cells were washed three times with blocking buffer and incubated with secondary HRP-conjugated goat anti-rabbit antibody for 1 h. Next, they were washed another three times with PBS and incubated with 100 µL of 3,3′,5,5′-tetramethylbenzidine solution (TMB, pre-warmed to room temperature, TCI Chemicals, Portland, OR, USA) for 30 min. To terminate the reaction, 100 µL of 1 N hydrochloric acid was added and the absorbance of each well was measured at 450 nm using a Synergy 4 microplate reader (BioTek Instruments, Winooski, VT, USA). The entire assay was performed at room temperature. 

### 4.8. Data Analysis

Calcium imaging data analyses were based on the amplitude of the intracellular calcium concentration and analyzed in Origin 9.6 (Version 9.6.0.172, OriginLab, Northampton, MA, USA). Statistical analysis was performed using either a one-sample *t*-test, unpaired Student’s *t*-test, or a one-way ANOVA with Tukey test for post hoc multiple comparisons in GraphPad Prism 9 (Version 9.5.0 (730), GraphPad Software, Boston, MA, USA) as appropriate and described in each experiment above. The level of significance was set at α = 0.05 for all experiments. All data are presented as mean ± SEM. 

## 5. Conclusions

The plasticity of the peripheral taste system is currently of great interest, and the taste responses originating in the oral cavity appear to be influenced by dietary experience and hormonal and nutritional status. Our present study in a human taste cell line indicates a potential effect of adiponectin signaling in the modulation of fat taste. We demonstrate that AdipoRon increases the translocation of CD36 to the plasma membrane of HuFF cells via the activation of AMPK and therefore selectively enhances their responses to fatty acids. Fat sensing is essential for the detection of essential fatty acids and the hormonal regulation of fat taste sensitivity may contribute to the regulation of fat intake in healthy subjects. Obese subjects display a diminished ability to detect fat concomitant with a much stronger preference for fat-rich foods [42,43]. Many studies have found plasma levels of adiponectin to be inversely correlated with body mass index, and high adiponectin levels correlate with a lower risk of diabetes [67]. Therefore, we speculate that the reduction of adiponectin levels in pathological states, such as obesity and diabetes, may contribute to alterations in the gustatory fat detection threshold, which, in turn, would affect fat intake. Considering the links between hormones, taste, fat intake, and obesity, understanding the mechanistic underpinnings of hormonal modulation of taste might present novel targets for appetite and weight control.

## Figures and Tables

**Figure 1 ijms-24-05801-f001:**
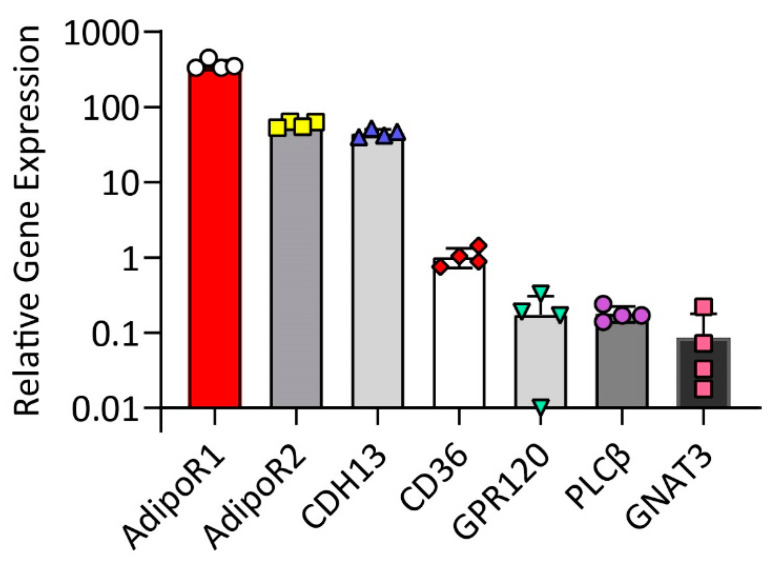
The relative mRNA expression of adiponectin receptors and fat taste signaling elements in HuFF cells as determined by quantitative real-time PCR using GAPDH as the housekeeping gene. Data from 3 replicates within four independent experiments are expressed as mean ± SEM relative to the expression of CD36.

**Figure 2 ijms-24-05801-f002:**
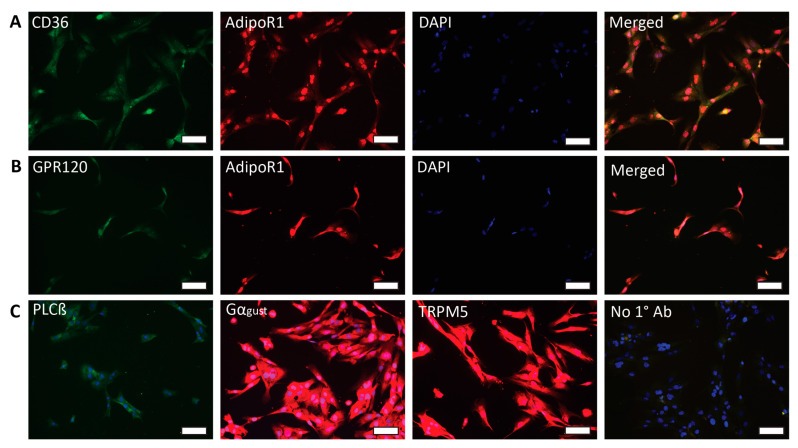
Localization of adiponectin receptor 1 (AdipoR1) and fatty acid transduction pathway elements in HuFF cells. (**A**), Double immunocytochemical labeling with antibodies against AdipoR1 (red) and CD36 (green) along with the nuclear stain DAPI (blue) reveal coexpression of both receptors in HuFF cells. (**B**), Labeling shows coexpression of GPR120 (green) and AdipoR1 (red) in HuFF cells. (**C**), HuFF cells also show significant positive labeling with antibodies against PLCß2, Gα_gust_, and TRPM5. Treatment with secondary antibodies with the omission of primary antibodies shows no positive immunoreactivity, only DAPI-stained nuclei are evident. Scale bars, 100 µM.

**Figure 3 ijms-24-05801-f003:**
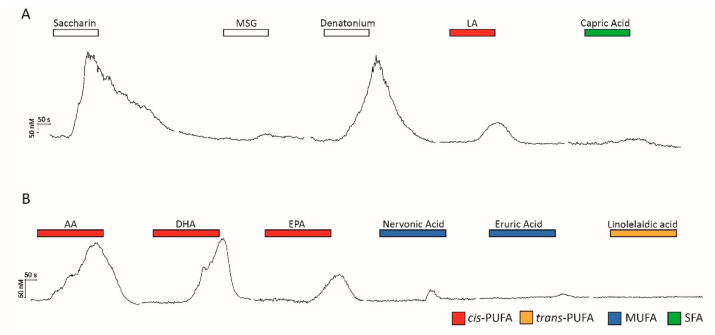
HuFF cells respond to a variety of tastants. (**A**), Saccharin (20 mM), denatonium (5 mM), linoleic acid (LA, 30 µM), and capric acid (100 µM) elicited reversible calcium responses in HuFF cells, but monosodium glutamate (MSG, 20 mM) did not reliably generate any changes in intracellular calcium in a significant proportion of HuFF cells (n = 280 total cells). Relative calcium response amplitudes are variable among cells. (**B**), HuFF cells responded to various *cis*-PUFAs (arachidonic acid, AA, ω-6, C20:4, 29 in 46 cells; docosahexaenoic acid, DHA, ω-3, C22:6, 22 in 46 cells; eicosapentaenoic acid, EPA, ω-3, C20:5, 35 in 46 cells), but not reliably to MUFAs (nervonic acid, NA, ω-9, C24:1, 2 in 46 cells; erucic acid, EA, ω-9, C22:1, 3 in 46 cells) or trans-PUFAs (linoelaidic acid, LEA, ω-6, *trans*-C18:2, 1 in 46 cells). The concentration of all fatty acids in this experiment was 10 µM.

**Figure 4 ijms-24-05801-f004:**
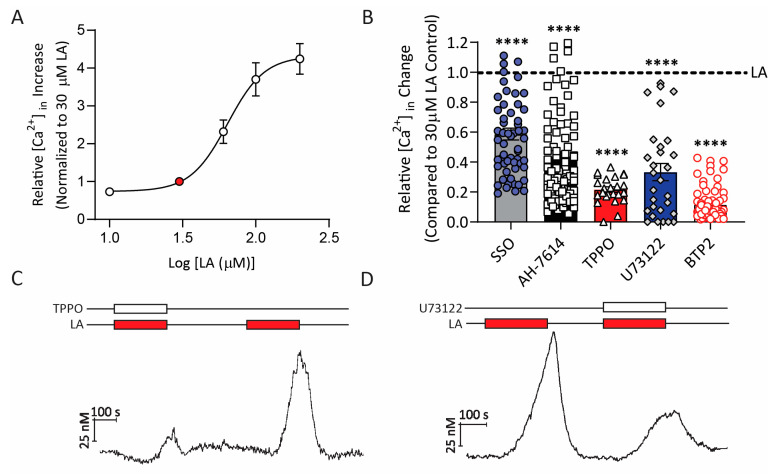
HuFF cells functionally express the PUFA-mediated transduction pathway. (**A**), Concentration–response curve for linoleic acid (LA) in HuFF cells. LA (10, 30, 60, 100, and 200 µM) was applied to Fura2-loaded HuFF cells during ratiometric calcium imaging assays. Data were normalized to the response to 30 µM LA within each cell, averaged, and presented as mean ± SEM. Data were fit by a logistic relation and the EC_50_ was determined to be 64.0 µM (*n* = 27–50 cells per point). (**B**), The intracellular calcium responses induced by LA (30 µM) were significantly reduced by sulfosuccinimidyl oleate (SSO, an irreversible inhibitor of CD36, 400 µM, 20 min pre-treatment), AH-7614 (GPR120 antagonist, 10 µM, 20 min pre-treatment), triphenylphosphine oxide (TPPO, a reversible and selective blocker for the TRPM5 channel, 200 µM, perfused with LA), U73122 (an aminosteroid phospholipase C inhibitor, 6 µM, perfused with LA), and BTP2 (a store-operated calcium entry blocker, 10 µM, 2 h pre-treatment). (**C**), Representative calcium trace showing responses to 30 µM LA with or without the TRPM5 inhibitor, TPPO, in HuFF cells. (**D**), Representative calcium trace showing responses to 30 µM LA with or without the PLC inhibitor, U73122, in HuFF cells. All data are presented as mean ± SEM. An unpaired Student’s *t*-test was used to determine statistical significance (**** *p* < 0.0001).

**Figure 5 ijms-24-05801-f005:**
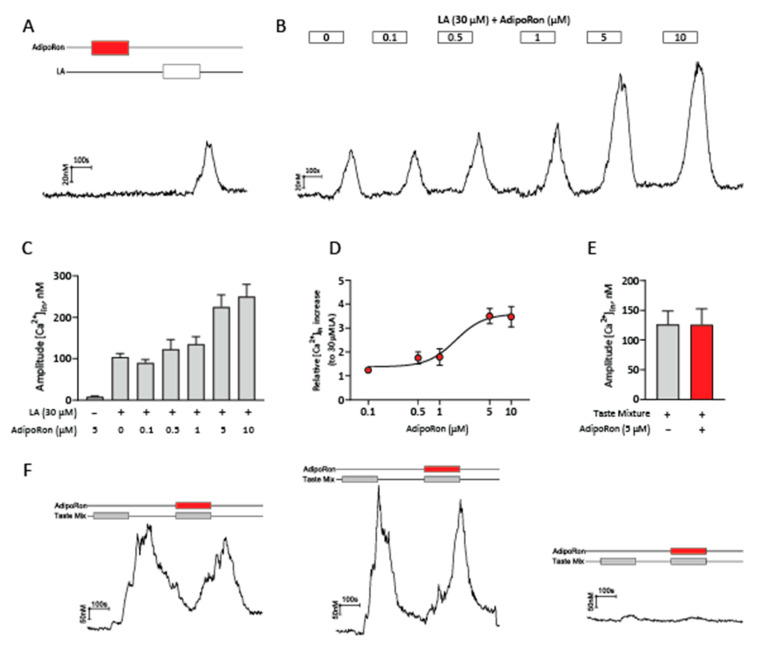
AdipoRon selectively enhances cellular responses to fatty acids (LA) in HuFF cells. (**A**), AdipoRon (5 µM) did not reliably generate any changes in intracellular calcium in HuFF cells. (**B**), Representative calcium trace showing responses to 30 µM LA with different concentrations (0, 0.1, 0.5, 1, 5, and 10 µM) of AdipoRon. (**C**), Mean amplitude measurements of responses to 30 µM LA with different concentrations of AdipoRon. (**D**), Dose–response curve for the enhancement effect of AdipoRon on LA-induced calcium responses in HuFF cells was plotted, fit with a logistic relation, and the EC_50_ value was determined to be 1.67 µM (*n* = 97–159 cells per point). (**E**), AdipoRon had no effect on taste mixture (sweet, bitter, and umami)-induced calcium responses (*n* = 51 cells). (**F**), Representative calcium trace showing responses to taste mixture with or without 5 µM AdipoRon in HuFF cells. Data are presented as mean ± SEM. An unpaired Student’s *t*-test was used to determine statistical significance.

**Figure 6 ijms-24-05801-f006:**
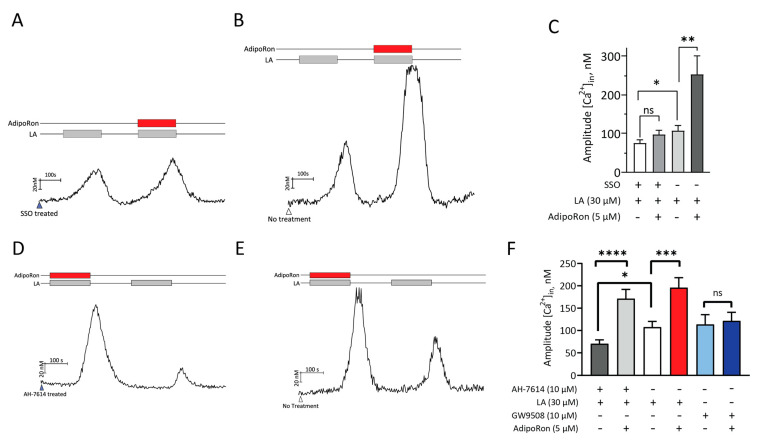
AdipoRon’s effect on the enhancement of fatty acid-induced responses is mediated by CD36, but not GPR120. Representative calcium traces showing responses to 30 µM LA with or without 5 µM AdipoRon in 400 µM SSO pre-treated (**A**) or untreated (**B**) HuFF cells. (**C**), Effect of SSO, AdipoRon, and SSO + AdipoRon on calcium responses to 30 µM LA. Blocking CD36 with SSO eliminated the enhancement role of AdipoRon on LA-induced calcium responses. Representative calcium traces showing responses to 30 µM LA with or without 5 µM AdipoRon in 10 µM AH-7614 (GPR120 antagonist) pre-treated (**D**) and untreated (**E**) HuFF cells. (**F**), Effect of AH-7614, AdipoRon, and AH-7614 + AdipoRon on calcium responses to 30 µM LA, and effect of AdipoRon on calcium responses to the GPR120 agonist GW9805 (10 µM). Blocking GPR120 did not affect the enhancement role of AdipoRon on LA-induced calcium responses, while AdipoRon had no effect on GW9508-induced calcium responses. Data are shown as mean ± SEM. Unpaired Student’s *t*-tests were used to determine statistical significance (ns *p* > 0.05, * *p* < 0.05, ** *p* < 0.01, *** *p* < 0.001, **** *p* < 0.0001).

**Figure 7 ijms-24-05801-f007:**
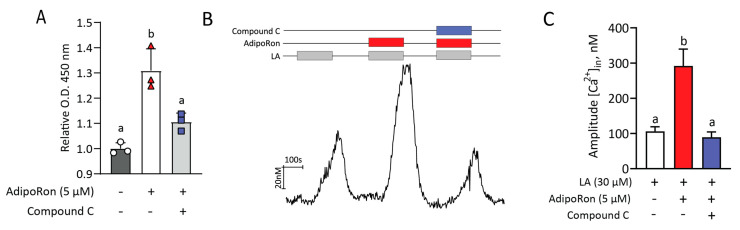
The effect of AdipoRon on LA-induced responses is dependent on AMPK activation. (**A**), Fold change in the phosphorylation of AMPKα (Thr172) induced by AdipoRon (5 µM) with and without compound C (Dorsomorphin, an AMPK inhibitor, 10 µM) relative to the control group at 30 min as measured by ELISA in three independent experiments. (**B**), Representative calcium trace showing responses to 30 µM LA with or without 5 µM AdipoRon and 10 µM compound C in HuFF cells. (**C**), Effect of AdipoRon and AdipoRon + compound C on calcium responses to 30 µM LA. Compound C inhibited the enhancement role of AdipoRon on LA-induced responses. Data are presented as mean ± SEM. A one-way ANOVA with the Tukey test for post hoc multiple comparisons was used to determine statistical significance. Letters above bars indicate statistical grouping.

**Figure 8 ijms-24-05801-f008:**
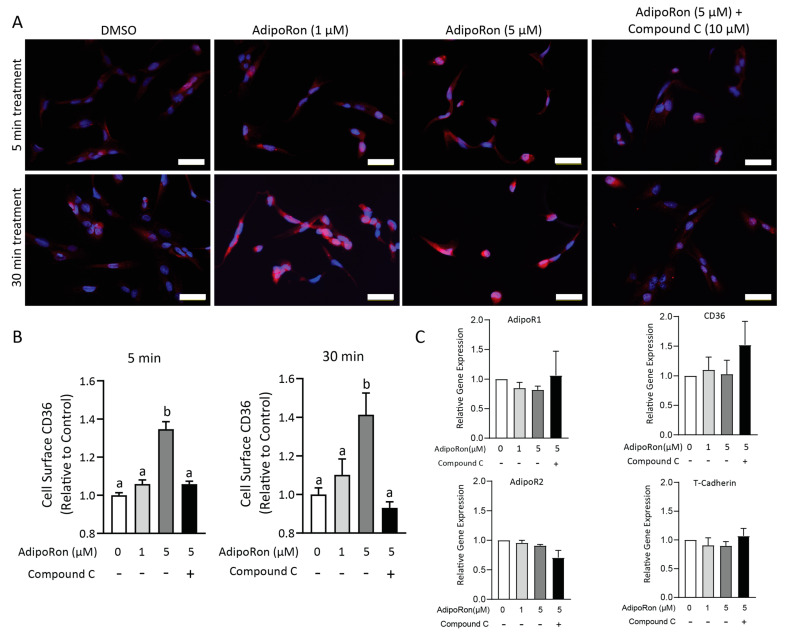
AdipoRon-induced CD36 subcellular translocation is inhibited by Compound C. (**A**), Representative images for the immunostaining of CD36 (red) along with the nuclear stain DAPI (blue) on HuFF cells, which were stimulated with 1 µM AdipoRon, 5 µM AdipoRon, 5 µM AdipoRon + 10 µM compound C, and DMSO (control) for 5 and 30 min, prior to fixation. Scale bars, 50 µM. (**B**), Fold change of cell surface CD36 relative to the control group by the CD36 translocation assay. (**C**), Relative gene expression of CD36 and adiponectin receptors (AdipoR1, Adipor2, and T-cadherin) measured by quantitative real-time PCR after treatment with AdipoRon with and without compound C. No significant differences in the gene expression were found between the stimulated HuFF cells and controls. Data are shown as mean ± SEM. A one-way ANOVA with the Tukey test for post hoc multiple comparisons was used to determine statistical significance. Letters above bars indicate statistical grouping.

## Data Availability

All relevant data are contained in the manuscript.

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
