# Peer review of "Adiponectin Enhances Fatty Acid Signaling in Human Taste Cells by Increasing Surface Expression of CD36"

_ijms, 2023, doi:10.3390/ijms24065801_

Round 1

Reviewer 1 Report

The impact of fat taste on human health is gaining attention. Although a lot of information is available for model animals, our knowledge on fat taste and relevant regulation by hormones is at its infancy for humans. This study by Lin et al. adds to our understanding of the molecular mechanisms underlying fat detection regulation in human taste cells. The researchers investigated the effect of adiponectin, a metabolic hormone released by fat tissue, on fatty acid responses using an immortalized cell line from human fungiform papillae (HuFF). They used a calcium imaging technique to visualize cellular responses as changes in intracellular calcium to achieve this goal. They also used immunocytochemistry and polymerase chain reaction (PCR) to supplement functional results.

The experiments were carefully designed and executed. The paper is well organized and well written. The data provide important new insights into how adiponectin modulates the detection of fatty acids by human taste cells.

I have a few criticisms that the authors should take into account.

Major points

 1. I was surprised the authors did not describe their experimental model in detail (HuFF). They should explain how human tissue is collected and the characteristics of human volunteers. Is the procedure for obtaining human fungiform taste cells the same as described by Ozdener et al. 2012 and 2014? Or did they create a new protocol? I believe they should provide more information on this subject.

2. Dietary lipids are primarily composed of triglycerides and not free fatty acids (FFAs). However, the authors used polyunsaturated fatty acids (PUFAs) to elicit taste responses in this study. Although lingual lipase efficiently hydrolyzes triglycerides in the oral cavity, releasing the taste stimulus (FFAs) in laboratory rodents, this is not the case in humans, as far as I am aware. As a result, the authors should address this caveat in their discussion.

3. The authors illustrated the presence of molecules involved in signal transduction in Type II cells in Figs. 1 and 2. However, only a small proportion of HuFF responded to taste stimuli that activate Type II transduction in functional tests (Supplementary Fig. 1). Indeed, according to Supplementary Fig. 1, the proportion of HuFF cells that respond to chemical stimuli is low. As a result, it is likely that many experiments were carried out on cells lacking the taste cell phenotype. This caveat should be discussed by the authors.

4. "Monosodium glutamate (MSG, 20 mM) did not reliably generate any changes in intracellular calcium in a significant proportion of HuFF cells," the authors write. It's strange that HuFF didn't react to monosodium glutamate. According to the presence of PLC, HuFF should be Type II cells (Fig. 2). Type II cells respond to umami stimuli as well. This limitation of their experimental model should be discussed by the authors.

5. There is an intense staining for AdipoR1 in the nuclei in Fig. 2A. Why is this so? Furthermore, according to Fig 2B, some cells appear to be polynucleated. Is it just an impression of mine?

6. In Fig. 4 the data presentation is perplexing. The authors first discuss the effect of inhibiting the CD36 and GPR120 receptors (Fig. 4D), and then discuss the effect on other signaling elements (Fig. 4B,C). I believe the data in the Figure should correspond to the logical presentation described in the text. Furthermore, I would anticipate calcium imaging data for the effect of SSO and AH-7614.

7. Data in Fig. 8: How do you know that CD36 was on the cell surface? The immunofluorescence signal appears to be in the cytoplasm to me. As it stands, I'm not convinced by this figure. The image quality is low, possibly because they are embedded in the text. To demonstrate CD36 translocation, I recommend focusing only on one cell at a higher magnification.

Minor points

Line 158: “Similar to responses shown in rodents,” please add reference

Fig. 5 E: Number of cells?

Line 281: Explain why you used a 30-min stimulation period. Is this time indicated in the commercially available kit?

Line 103, 107, and Fig. 1: PLCβ? Do they mean PLCβ2?

Author Response

REVIEWER 1:

1A. I was surprised the authors did not describe their experimental model in detail (HuFF). They should explain how human tissue is collected and the characteristics of human volunteers. Is the procedure for obtaining human fungiform taste cells the same as described by Ozdener et al. 2012 and 2014?...

We provide more detail in the current version to indicate more clearly that this human fungiform cell line is a commercially available cell line from Applied Biological Materials (Catalog # T0029).  Since to our knowledge this is one of the first manuscripts to use this cell line, we have spent time and effort to validate its utility for studies of fatty acid signaling and show it has all the cellular machinery needed for studying polyunsaturated fatty acid transduction and its regulation by adiponectin.

1B. Dietary lipids are primarily composed of triglycerides and not free fatty acids (FFAs). However, the authors used polyunsaturated fatty acids (PUFAs) to elicit taste responses in this study. Although lingual lipase efficiently hydrolyzes triglycerides in the oral cavity, releasing the taste stimulus (FFAs) in laboratory rodents, this is not the case in humans, as far as I am aware. As a result, the authors should address this caveat in their discussion.

This has long been controversial in the field. While it is true that there are reports of limited human lingual lipase activity, there are also many publications that indicate that support the existence of lipase activity in human saliva leading to a partial triglyceride breakdown of FFA (Neyraud et al, 2012; Voigt et al, 2014) and humans clearly appear to able to detect FFAs (Running and Mattes, 2014; Running et al, 2015). Moreover, it has been recognized that the concentration of free fatty acids in many fat-containing foods are at or well above the concentrations that are known to activate the GPR120/CD36 dependent pathways examined in the current study. 

1C. The authors illustrated the presence of molecules involved in signal transduction in Type II cells in Figs. 1 and 2. However, only a small proportion of HuFF responded to taste stimuli that activate Type II transduction in functional tests (Supplementary Fig. 1). Indeed, according to Supplementary Fig. 1, the proportion of HuFF cells that respond to chemical stimuli is low. As a result, it is likely that many experiments were carried out on cells lacking the taste cell phenotype. This caveat should be discussed by the authors.

The HuFF cells represented an immortalized cell line derived from human fungiform taste cells and the identity of the founding cell(s) subtypes are not known.  We agree that it has many of the hallmarks of Type II cells making it suitable for studies of taste transduction involving those tastants that activate this cell type.  As the manuscript shows, it appears particularly well suited for studies in fatty acid (fat) transduction and may be suitable for other taste modalities as well.  Of course, the latter point would also need further investigation.  While this is an immortalized cell line, it contains cells in all stages of development from those just being ‘born’, through those developing more ‘adult’ phenotypes to those going through senescence and eventually apoptosis and death.  In taste we do not have a thorough understanding of the chemosensory abilities of individual cells as they go through these stages. It is feasible that as they ‘age’ they acquire and/or lose specific chemosensory phenotypes giving the impression the group is more heterogenous than in actuality.  As well, there may be conditions in the culturing of the cells that favor the expression of certain receptor profiles over others as these cells grow and develop that, for example, may negatively impact expression of umami receptors.  We simply do not understand this process well enough to state definitively.  Nonetheless, we show data that does support the use of these cells for studies of fatty acid transduction and its modulation by adiponectin.

No non-responsive cells were included in any analysis regarding the effects of fatty acids and their modulation by AdipoRon. We merely included the proportion of cell responses to help validate the suitability of HuFF cells for studies of signaling in taste.

1D. Monosodium glutamate (MSG, 20 mM) did not reliably generate any changes in intracellular calcium in a significant proportion of HuFF cells," the authors write. It's strange that HuFF didn't react to monosodium glutamate. According to the presence of PLC, HuFF should be Type II cells (Fig. 2). Type II cells respond to umami stimuli as well. This limitation of their experimental model should be discussed by the authors.

Please see our Discussion in the first paragraph of response 1C above as this applies here.  We have added some information in the Discussion on page 11 (line 373 and following) to clarify this point.

1E. There is an intense staining for AdipoR1 in the nuclei in Fig. 2A. Why is this so? Furthermore, according to Fig 2B, some cells appear to be polynucleated. Is it just an impression of mine?

We agree the staining looks intense for AdipoR1 which we attribute to the fact that this is highly expressed in taste cells according to scRNAseq studies compared to most taste genes (Sukumaran et al. Sci Rep 2017, 7, 7595). Previous data have shown that AdipoR1 fluorescent signals accumulate in close apposition to the plasma membrane as well as in an intracellular compartment surrounding the nucleus in other types of cells (Almabouada et al, 2013; Jung et al, 2017). We do not have an insightful answer as to why Fig. 2A looks to be more intense in the nuclei. We also interpret the apparent “multinuclear’ cells to be cells that are in close apposition to one another or even perhaps in the initial stages of mitosis. This is not something we have noticed with other antibodies and these cells and may be reflective of the intensity of staining as well.    

1F. In Fig. 4 the data presentation is perplexing. The authors first discuss the effect of inhibiting the CD36 and GPR120 receptors (Fig. 4D), and then discuss the effect on other signaling elements (Fig. 4B,C). I believe the data in the Figure should correspond to the logical presentation described in the text. Furthermore, I would anticipate calcium imaging data for the effect of SSO and AH-7614.

We agree and thank you for the comment. We apologize that it is unclear and reflects a holdover from an earlier version of the manuscript, where we had included representative traces for SSO and AH-7614 here, but were later deleted to conserve space as similar data are shown in Fig. 6. Nonetheless, to make it match the descriptions in the text chronologically we have reordered the graphs in Fig. 4.

1G. Data in Fig. 8: How do you know that CD36 was on the cell surface? The immunofluorescence signal appears to be in the cytoplasm to me. As it stands, I'm not convinced by this figure. The image quality is low, possibly because they are embedded in the text. To demonstrate CD36 translocation, I recommend focusing only on one cell at a higher magnification.

This is a very good point. Using immunofluorescence alone, we agree with the reviewer it is not possible to conclude definitively that the change is entirely on the cell surface due to the limits of the this immunofluorescent assay. Nonetheless our images clearly show there are changes in the intensity of the fluorescent signal near the plasma membrane area with higher intensity in the AdipoRon treated group that is reduced in the Dorsomorphin (Compound C) treated group, consistent with our conclusions. Frankly, this is the reason we used an additional technique, the CD36 translocation assay (Fig. 8B) that measures cell surface CD36 (Samovski et al., 2012).  This, too, was in line with our conclusions regarding the effect of AdipoRon on fatty acid signaling.

1H. Line 158: “Similar to responses shown in rodents,” please add reference.

This has been added.

1I. Fig. 5 E: Number of cells?

The 51 cells responding to taste mixture are included and this information has been added to the figure legend.

1J. Explain why you used a 30-min stimulation period. Is this time indicated in the commercially available kit?

This was more of a practical concern.  We had used 5 and 30 minutes in other experiments included in the paper and the ELISA kit, which was expensive, only permitted us to design an experiment with three technical replicates (experiments) at one time point. We chose 30 minutes to ensure that the AdipoRon treatment was sufficient to record any AdipoRon-dependent changes. Ideally, we would have liked to done multiple time points, but the data nonetheless agree with the conclusions drawn in the paper from multiple types of data.

1K. Line 103, 107, and Fig. 1: PLCβ? Do they mean PLCβ2?

Apologies for the typo.  Yes, it is PLCß2. We have corrected the text.

Reviewer 2 Report

The authors studied the effect of AdipoRon (an adiponectin receptor agonist) on fatty acid-induced calcium signaling. Experiments were performed on a human taste cell line (Huff cells). The work is very interesting and novel. The topic of the interaction of adiponectin signalling with signalling via CD36 opens new horizons in obesity research.

Overall, the experimets are well done, the results are clearly commented and discussed.

I have only minor comments:

1) Why is mRNA expression in Figure 1 related to CD36? In my opinion it would be more relevant to show the "raw" data as 2-dCT

2) Why in some figures/experiments the effect of LA (linoleic acid) is shown first and then the combined effect of LA+inhibitor/AdipoRon, and in other figures first LA+inhibitor and then LA alone is shown (see Figure 4 B x C; 6 -A,B x D, E). Is the response somehow different according to the order of the factors given?

3) In the description of some graphs, the number of samples on the basis of which the graph was created and which were included in the statistical analysis is missing (specifically Fig. 5C, E; 6 C, F; 7C; 8B,C). Graphs with individual values could also be used, as in Figure 1.

4) On the line 175-177 the sentence:"Treatment with…." doesn't make sense. There seems to be a missing verb. Please rephrase.

Author Response

Please see attached letter indicating all responses to reviewers. Thank you. 

Reviewer 3 Report

Dietary fat intake is positively correlated with BMI, and the gustatory response to fatty acid consumption is reduced in obesity. Adiponectin has been shown to be involved in fatty acid detection, but its specific role and mechanism of action in the gustatory system are unclear. This study investigated the impact of adiponectin receptor signaling on calcium responses to fatty acids and other tastants. Using different inhibitors of gustatory signaling pathways, this study showed that the adiponectin receptor agonist, AdipoRon, activates AMPK to stimulate CD36 translocation to the cell membrane, resulting in an enhanced response to fatty acids in HuFF cells. Additionally, the researchers first characterized the recently developed HuFF cell line, and demonstrated that they function similarly to rodent taste cells, which have been well characterized. The studies are clearly described. The conclusions are accurately based on the study’s results. A few minor comments can be found below.

·         The sentence starting “Additionally, behavioral studies…” in lines 63-65 appears to be missing a word.

·         Check that appropriate statistical tests were being used. Unpaired t-tests do not appear to be the appropriate tests.

Author Response

(The authors gave the same response as above.)

Round 2

Reviewer 1 Report

All of my criticism was satisfactorily handled by the authors.